# Numerical Simulation of Rock Mass Structure Effect on Tunnel Smooth Blasting Quality: A Case Study

**Jianxiu Wang** [1,2,3,*], **Ansheng Cao** [1], **Jiaxing Liu** [1], **Huanran Wang** [3], **Xiaotian Liu** [1], **Huboqiang Li** [1], **Yuanwei Sun** [1], **Yanxia Long** [1] and **Fan Wu** [1]

1   College of Civil Engineering, Tongji University, Shanghai 200092, China; 2110410@tongji.edu.cn (A.C.); dj12ljx@tjad.cn (J.L.); 2012liuxiaotian@tongji.edu.cn (X.L.); 2030168@tongji.edu.cn (H.L.); 2132198@tongji.edu.cn (Y.S.); 2010328@tongji.edu.cn (Y.L.); 2030167@tongji.edu.cn (F.W.)
2   Key Laboratory of Geotechnical and Underground Engineering of Ministry of Education, Tongji University, Shanghai 200092, China
3   Key Laboratory of Impact and Safety Engineering, Ministry of Education, Ningbo University, Ningbo 315211, China; wanghuanran@nbu.edu.cn
*   Correspondence: wangjianxiu@tongji.edu.cn; Tel.: +86-139-16185056 or +86-21-65983036; Fax: +86-21-65985210

**Abstract:** Taking the Zigaojian tunnel, Hangzhou–Huangshan high-speed railway, China, as background, the rock mass structure effect on smooth blasting quality was studied. Four rock mass structures were determined on the basis of the information collected on the tunnel site. Smooth blasting finite element models were established using LS-DYNA. The accuracy of the numerical calculation model was verified by comparing the overbreak and underbreak between the numerical simulation and monitoring. Orthogonal numerical test was used to study the rock mass structure effect through single factor and main effect analysis methods. With the decrease in rock mass integrity, the smooth blasting overbreak of tunnels with massive integrity structure, massive structure, layered structure, and cataclastic structure increased. For massive integrity structure and cataclastic structure, the peripheral hole spacing should be emphatically considered. Meanwhile, in massive structure and layered structure, the included angle and spacing of structural planes had a great influence on the smooth blasting quality. The research results could provide a reference to improve the quality of similar tunnel smooth blasting.

**Keywords:** tunnel smooth blasting; rock mass structure effect; overbreak and underbreak; field test; orthogonal numerical test

## 1. Introduction

With the acceleration of urbanization, tunnel construction projects are recently increasing in China. In the construction of tunnel engineering, the contour and size of the excavation section should be accurate, and the disturbance to surrounding rock should be small. Smooth blasting has been widely used in tunnel blasting construction because of its good controlled blasting profile and small disturbance to surrounding rock [1–3]. However, a large number of structural planes and structural bodies in the rock mass exist, which are arranged and combined in different manners to form different rock mass structures; rock mass structure is also an important cause of overbreak and underbreak of smooth blasting [4–6]. At present, rock mass structure is mainly divided into five types: massive integrity structure, massive structure, layered structure, cataclastic structure, and granular structure [7,8]. The overbreak and underbreak in smooth blasting affect the work efficiency of subsequent processes, such as ballast transportation and tunnel maintenance, and increase the cost of tunnel excavation. Therefore, research on the effect of rock mass structure in the smooth blasting of tunnels has become an indispensable part of scientific research and production.

Many factors affect the quality of tunnel smooth blasting, which is difficult to control. How to correctly and reasonably control the quality of tunnel smooth blasting has attracted the attention of researchers, and fruitful studies have been carried out. During the construction of the Fuling highway long tunnel, Liu et al. [9] conducted 18 field smooth blasting tests and studied the influence of rock-saturated uniaxial compressive strength, peripheral hole spacing, pressure relief hole spacing, the minimum charge amount of peripheral holes, and linear charge concentration on the quality of tunnel smooth blasting. On the basis of the three-dimensional blasting model of LS-DYNA, Zou et al. [10] simulated tunnel smooth blasting. Meanwhile, the orthogonal test design method was used to study the effects of seven factors on the over- and under-excavation of the tunnel, including peripheral hole spacing, least resistance line, charge density, charge form, rock mass type, detonation speed, and borehole inclination. The study revealed that the rock mass type has the greatest effect on the blasting quality, while the charge density and detonation velocity could be regarded as secondary factors under specific site conditions. On the basis of model tests and field investigation, Chakraborty et al. [11,12] evaluated the joint direction and rock mass quality affecting tunnel overbreak and considered that blasting quality was related to the average rock mass size, depth, and geometric dimensions of the cross-sectional profile. Mei et al. [13] used the horizontal layered rock mass large-section tunnel as the research object and proposed the cutting methods of "center holes and four-wedge cutting holes", "empty holes, long holes, short holes, and additional relief holes", and the fine excavation mode of the maximum single hole charging scheme. Meng et al. [14] improved smooth blasting parameters and construction technology in large dip tunnels in accordance with the actual construction situation on site, the rock, and the explosive properties. In smooth blasting, Kim et al. [15] proposed a controlled blasting method using pilot holes to smooth the smooth fracture surface and reduce the blasting damage zone. Gao et al. [16] studied the blasting method of slotted tube charging structure in fault stratum through a field test. The research showed that the circumferential hole with a slotted tube charging structure significantly improved the overbreak phenomenon in the fault stratum. Yang et al. [17] adopted self-made uniform similarity materials and digital image correlation methods and found that with the increase in buried depth, the ground stress exhibited a significant effect on smooth blasting, and the fracture surface tended to be flat with the increase in in situ stress. Qi et al. [18] used the finite element numerical simulation software ANSYS/LS-DYNA to simulate the blasting process of the coupled charge in the peripheral holes in smooth blasting. The results showed that when the peripheral hole spacing was 0.6 m, the surrounding rock stress and spalling were the smallest, which could improve the efficiency of smooth blasting.

Besides the studies in field tests, laboratory tests and numerical simulation on the quality improvement of smooth blasting, over- and under-excavation prediction model, random model, and comprehensive optimization control model were used to select the best blasting scheme. Taking the smooth blasting of Koyna Lake Tap Tunnel in India as an example, Murthy et al. [19] compared the prediction model calculated by the peak particle velocity (PPV) of blasting particles with the measured overbreak and underbreak. They proposed the overbreak and underbreak BIRD prediction model and calculated the crack development state and the PPV threshold of overbreak and underbreak. Thereafter, Dey and Murthy et al. [20] adopted the BIRD prediction method on the basis of the measured data of the tunnel to control the overbreak and underbreak errors of the four tunnels within 10%, thus verifying the feasibility of the BIRD overbreak and underbreak prediction model. Monjezi et al. [21] proposed a prediction model of overbreak and underbreak on the basis of fuzzy set theory, which controlled overbreak and underbreak by adjusting the four parameters of blockage length, hole depth, charge amount, and hole spacing. Sari et al. [22] established a Monte Carlo stochastic model on the basis of controllable parameter adjustment to predict overbreak and underbreak. Zou [23] et al. proposed the constructing method of tunnel smooth blasting quality control index system and established the smooth blasting quality control index system with levels of geological

conditions, explosive properties, borehole parameters, charging parameters, initiation method, tunnel parameters, and construction factors as indices.

The predecessors mainly considered the physical and mechanical properties of rock mass, explosive properties, charging parameters, and construction methods in researching the quality of tunnel smooth blasting. However, the influence of the types of rock mass structure on tunnel overbreak and underbreak is not fully considered. Besides, the current research on the influence of rock mass structure on the overbreak and underbreak of tunnel smooth blasting is mostly focused on a single rock mass structure; it rarely involves different rock mass structures. Therefore, on the basis of field tests, this paper adopted the method of orthogonal numerical tests to study the influence of the rock mass structures, such as massive integrity structure, massive structure, layered structure, and cataclastic structure, on the quality of tunnel smooth blasting.

## 2. Material and Methods

### 2.1. Background

The Zigaojian tunnel, Hangzhou–Huangshan Railway, was selected as the background, and the sections DK 149 + 928–DK 149 + 957, DK 150 + 876–DK 150 + 895, and DK 152 + 847–DK 152 + 902 were selected as the test sections and monitoring objects, as shown in Figure 1.

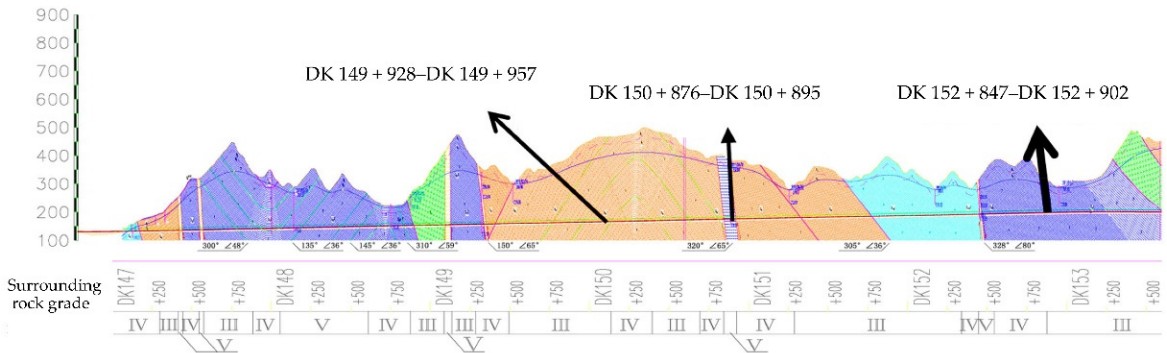

**Figure 1.** Geological longitudinal section of Zigaojian tunnel (unit: m).

The surrounding rock grade of the monitoring section DK 149 + 928–DK 149 + 957 was grade III, and the lithology was fine sandstone, belonging to relatively soft rock, with a rock density of 2.6 g/cm$^3$. The monitoring section DK 150 + 876–DK 150 + 895 was located in the fault fracture zone, and the surrounding rock grade was grade V. The joints at this section were well developed. The rock mass was broken. The lithology was siltstone intercalated with silty mudstone, and the rock hardness was soft rock. The density of the rock was 2.0 g/cm$^3$. The rock lithology of the monitoring section DK 152 + 847–DK 152 + 902 was lithic sandstone, which belonged to hard rock. The surrounding rock grade included grades III and IV sections, and the rock density was 2.6 g/cm$^3$.

### 2.2. Rock Mass Structure Analysis and Mechanical Information

Information on the structural surface of the tunnel face was collected using manual collection, three-dimensional scanners, and infrared thermal imaging cameras onsite. The occurrence and spacing of structural planes, which were easy to identify, were directly measured by geological compass. The infrared thermal imager was used for auxiliary discrimination of discontinuities that were difficult to identify because of dust cover and weathering zone. For the structural planes beyond the manual measurement range, the three-dimensional scanner was used for omnidirectional scanning, and the results were imported into the computer for three-dimensional data reorganization. Geomagic Studio Software was used for post-processing of the scanning results. After virtual imaging was conducted, the occurrence was measured. The uniaxial compressive and tensile strength of

rocks were measured by point load test. Through field measurement of rock mass structure information and structural plane statistics, the monitoring section DK 149 + 928–DK 149 + 957 was found to have a set of controllable structural planes of 157° ∠62°, with an average spacing of 3 m, which is marked with a red line in Figure 2a.

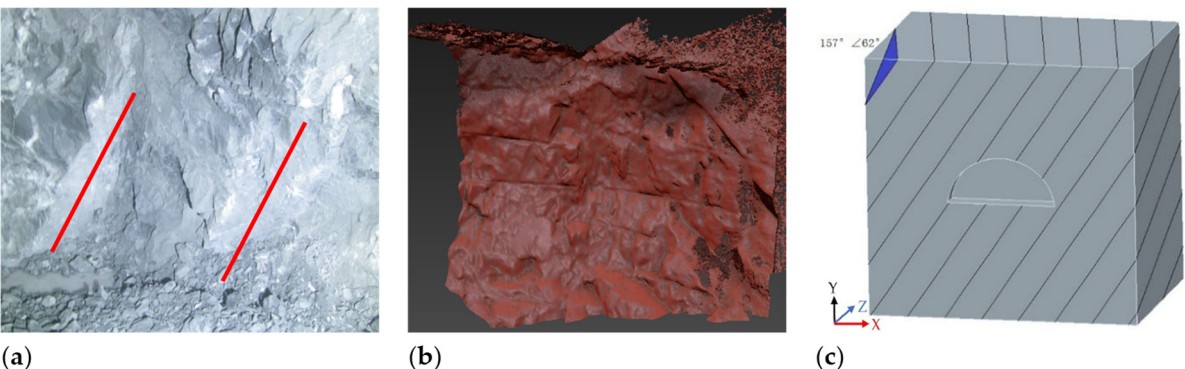

**Figure 2.** Monitoring section DK 149 + 928–DK 149 + 957 structural surface information: (**a**) visible light photo of the rock wall; (**b**) 3D scan of the rock wall; and (**c**) geometric model.

The rock mass structure type was a massive integrity structure. The three-dimensional scanning pictures and geometric model diagrams are shown in Figure 2b,c. The average tensile strength and compressive strength of rocks were 2.37 and 44.57 MPa, respectively, as measured by point load test.

Many and staggering structural planes were found in DK 150 + 876–DK 150 + 895. The rock mass was very broken, no control structural plane was observed, and the rock mass was a cataclastic structure. The structural plane information is shown in Figure 3a,b. The average tensile strength of the rock was 1.53 MPa, and the average compressive strength was 32.17 MPa. Due to the extremely developed structural planes in the fragmented structure, it could be considered as a continuous medium in the numerical simulation. The strength of the rock mass was reduced based on the rock strength. The geometric model was the same as the complete rock mass structure, as shown in Figure 3c. Starting from the 1970s, many scholars have proposed some treatment methods for reducing rock mass strength when jointed rock mass was considered as an equivalent continuous medium. In 1993, Aydan [24] proposed using elastic longitudinal wave velocity to estimate the uniaxial compressive strength of weak rock masses. In 1995, Barton [25] et al. proposed a formula for estimating the uniaxial compressive strength of rock mass by using its longitudinal wave velocity. In 2005, Singh and Rao [26] proposed an empirical method for estimating rock mass strength on the basis of deformation modulus. The strength reduction coefficient of the above empirical determination method of the equivalent strength of jointed rock mass was based on wave velocity or elastic modulus, but it was not satisfactory from the perspective of engineering application, and the dispersion of the estimation results was large. The reason may be that these evaluation standards were not fully considered, and the evaluation factors were greatly affected by the occurrence of environmental conditions of the tested rock mass. Thus, in recent years, some scholars proposed a method to determine the equivalent continuum strength on the basis of rock mass classification index, and they achieved good results. In 1997, Hock and Brown [27] proposed the uniaxial compressive strength formula of rock mass that was based on the study of a large number of complete rock brittle failure test data and many jointed rock mass characteristic models. In 1995, Kalamaras and Bieniawski [28] proposed an estimation formula of rock mass equivalent strength that was based on the RMR rock mass classification index. In 2002, Barton [29] established the uniaxial compressive strength formula of rock mass in accordance with Q classification as follows:

$$\sigma_{cm} = 5\gamma(Q\sigma_{ci}/100)^{1/3}, \tag{1}$$

where $\sigma_{cm}$ is the uniaxial compressive strength of the rock mass (MPa), $\sigma_{ci}$ is the uniaxial compressive strength of the rock (MPa), $\gamma$ is the rock density (g/cm$^3$), and $Q$ is the classification value of rock mass.

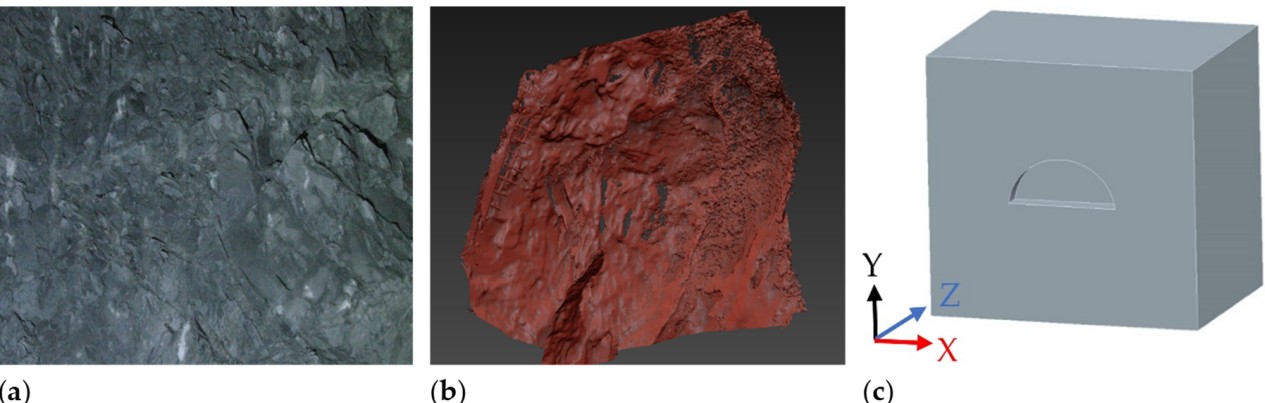

**Figure 3.** Monitoring section DK 150 + 876–DK 150 + 895 structural surface information: (**a**) visible light photo of the rock wall; (**b**) 3D scan of the rock wall; and (**c**) geometric model.

From the engineering example, the calculation results of the rock mass equivalent continuous medium strength by using the rock mass classification system of $Q$ classification were closer to reality than those of wave velocity and deformation modulus conversion method. Consequently, the equivalent strength of cataclastic rock mass was reduced using the equivalent continuous medium-strength conversion formula of rock mass that was based on $Q$ proposed by Barton, and the $Q$ was 0.6.

The surrounding rock grade of the DK 152 + 847–DK 152 + 902 monitoring section included grades III and IV, and the mileages were DK 152 + 847–DK 152 + 875 and DK 152 + 875–DK 152 + 902, respectively. In accordance with the statistical results of the structural planes of DK 152 + 847–DK 152 + 875, two groups of control structural planes were identified in the rock mass, namely, the average attitude was 308°∠57°, with an average spacing of 1.9 m, and an average attitude was 287°∠38°, with an average spacing of 2.9 m. The rock mass was found to be a massive rock mass structure, as shown in Figure 4.

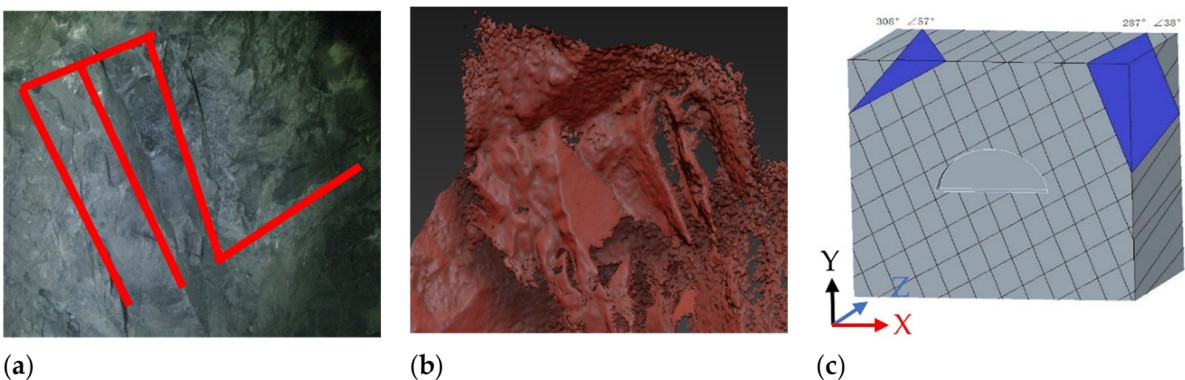

**Figure 4.** Monitoring section DK 152 + 847–DK 152 + 875 structural surface information: (**a**) visible light photo of the rock wall; (**b**) 3D scan of the rock wall; (**c**) geometric model.

The rock mass in DK 152 + 875–DK 152 + 902 had a set of controlled structural planes, with an average occurrence of 318°∠52° and an average spacing of 1.5 m, which was a layered rock mass structure, as shown in Figure 5.

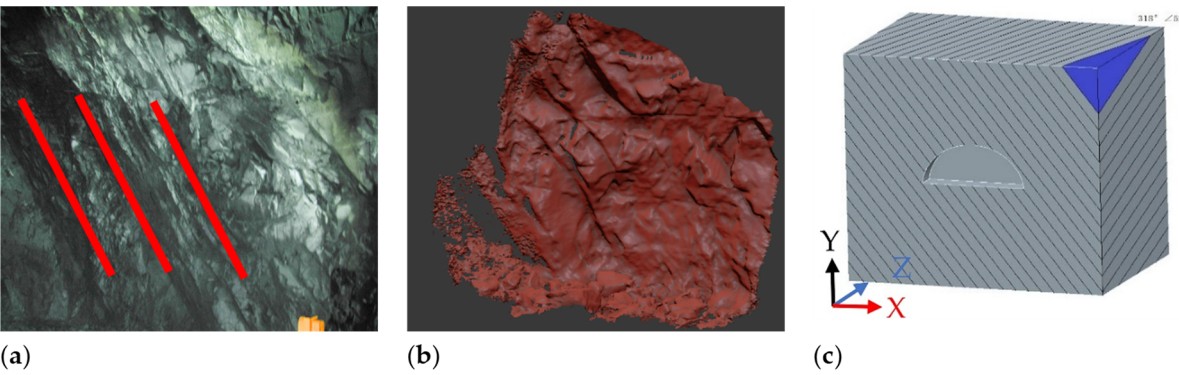

(**a**)                                        (**b**)                                        (**c**)

**Figure 5.** Monitoring section DK 152 + 875–DK 152 + 902 structural surface information: (**a**) visible light photo of the rock wall; (**b**) 3D scan of the rock wall; and (**c**) geometric model.

### 2.3. Blasting Scheme and Monitoring

The blasting scheme of the monitoring section adopted the step method, which was divided into two parts: upper step and lower step. As only the upper step was blasted during the test, the numerical model only studied the upper step. The section height of the upper step was 6.26 m, the width was 13.00 m, and the area was 63.48 m$^2$. The blast hole layout is shown in Figure 6.

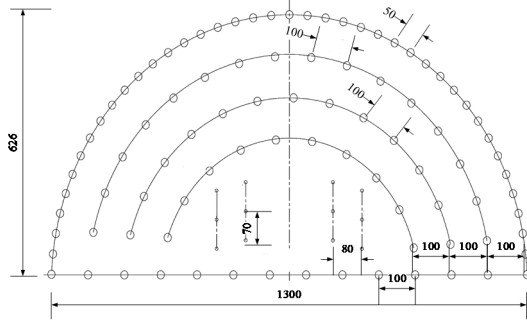

**Figure 6.** Layout of the blast hole (unit: cm).

At the tunnel site, the ZTSD-3 tunnel section instrument was used to measure the overbreak and underbreak of the smooth blasting tunnel face. In accordance with the field test results, the average overbreak area, average underbreak area, and average maximum overbreak of the tunnel in each monitoring section are plotted in Figure 7.

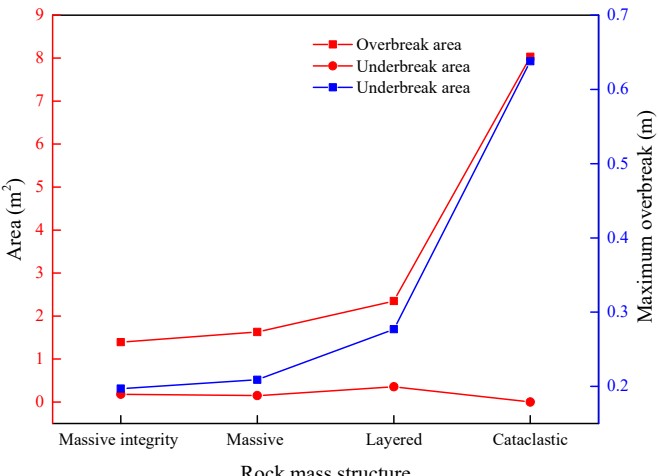

**Figure 7.** Relationship between overbreak and underbreak and rock mass structure types.

The integrity of massive integrity structure, massive structure, layered structure, and cataclastic structure decreased successively. Figure 7 shows that the average overbreak area and the average maximum overbreak increased with the decrease in rock mass integrity, especially for cataclastic structures, whose overbreak area and maximum overbreak were much larger than other structural types. Therefore, in the smooth blasting of a cataclastic rock tunnel, appropriately increasing the peripheral hole spacing or reducing the charge was necessary. The relationship between the average underbreak area and the type of rock mass structure was not obvious, and the underbreak area was only approximately one-tenth of the overbreak area, which was not significant.

### 2.4. Mathematical Model and Numerical Mode

#### 2.4.1. Surrounding Rock Mathematical Model

The Johnson–Holmquist–Cook (JHC) material constitutive model was first proposed by Johnson et al. [30,31] in 1993. It was a rate-dependent material constitutive model that can better describe the material in large strain and high strain; dynamic response under high stress conditions was initially used to study the problem of concrete penetration and then gradually applied to the study of the dynamic response of rocks [32–34]. Therefore, the JHC concrete constitutive model was selected to simulate surrounding rock in the present paper. The parameters for surrounding rock model calculation included static compressive strength $f_c$; density $\rho_0$; strength parameters $A$, $B$, $C$, $N$, $S$max, and $G$; damage parameters $D_1$, $D_2$, and *EFMIN*; and pressure parameters $P_c$, $\mu_c$, $P_1$, $\mu$L, $K_1$, $K_2$, $K_3$, and $T$. The above 19 parameters, together with the reference strain rate $\dot{\varepsilon}_0$ and failure type $f_s$, constituted all 21 calculation parameters of the JHC model. The density, elastic modulus, compressive strength, and other properties of surrounding rock were obtained through field tests. Referring to the original literature [31] and the JHC parameter calculation method for surrounding rock, the remaining parameters were approximately calculated, the numerical simulation results were inversed with the field monitoring results, and the JHC model parameters were adjusted to make the calculation results consistent with the field blasting results. The material parameters of the JHC model were determined, as shown in Table 1.

**Table 1.** JHC model parameters of rock mass (uint: g-cm-μs).

| $\rho_0$ | $G$ | $A$ | $B$ | $C$ | $N$ | $f_c$ |
|---|---|---|---|---|---|---|
| Field measurements | 0.132 | 0.79 | 1.60 | 0.007 | 0.61 | Field measurements |
| $T$ | *EPS0* | *EFMIN* | *SFMAX* | $P_c$ | $\mu_c$ | $P_l$ |
| Field measurements | $1.0 \times 10^{-6}$ | 0.01 | 7 | $1.08 \times 10^{-4}$ | $7.18 \times 10^{-4}$ | $1.05 \times 10^{-2}$ |
| $\mu_1$ | $D_1$ | $D_2$ | $K_1$ | $K_2$ | $K_3$ | $f_s$ |
| 0.1 | 0.01 | 1.00 | 0.174 | 0.388 | 0.2988 | −0.004 |

#### 2.4.2. Explosive Material Model

The high explosive model *MAT_HIGH_EXPLOSIVE_BURN [30] was used to simulate the explosive material, and the JWL equation of state was used to simulate the relationship between the pressure and the specific volume in the explosive detonation process as follows:

$$P = A\left(1 - \frac{\omega}{R_1 V}\right)e^{-R_1 V} + B\left(1 - \frac{\omega}{R_2 V}\right)e^{-R_2 V} + \frac{\omega E_0}{V}, \tag{2}$$

where $P$ is the detonation pressure; $V$ is the relative volume; $E_0$ is the initial specific internal energy; and $A$, $B$, $R_1$, $R_2$, and $\omega$ are material constants. In this paper, No. 2 rock emulsion explosive was used, and its materials and state equation parameters are shown in Table 2.

| $\rho$ (g/cm$^3$) | $v_0$ (cm/$\mu$s) | $PCJ$ ($10^5$ MPa) | $A$ ($10^5$ MPa) | $B$ | $R_1$ | $R_2$ | $\omega$ | $E_0$ ($10^{-1}$ J/cm$^3$) |
|---|---|---|---|---|---|---|---|---|
| 1.0 | 0.45 | 0.05 | 524.2 | 0.769 | 4.2 | 1.0 | 0.3 | 8.5 |

### 2.4.3. Numerical Model and Verification

The finite element software LS-DYNA was used to establish the smooth blasting finite element model of the tunnel, whose rock mass structure was massive integrity structure, massive structure, layered structure, and cataclastic structure, by referring to the information of structural plane collected onsite, as shown in Figure 8. The dimension of the model along the X, Y, and Z directions was 80 m $\times$ 80 m $\times$ 60 m. The tunnel was 6.26 m high and 13.00 m wide. Except for the excavated part, the boundary surface of the model adopted the non-reflection boundary condition. The non-reflective boundary condition was defined by the keyword *BOUNDARY_NON_REFLECTING in LS-DYNA. The *CONTACT_SURFACE_TO_SURFACE command [30] in LS-DYNA was used to simulate the contact structural plane for the two adjacent blocks not to penetrate each other and to transmit normal pressure and tangential friction to each other.

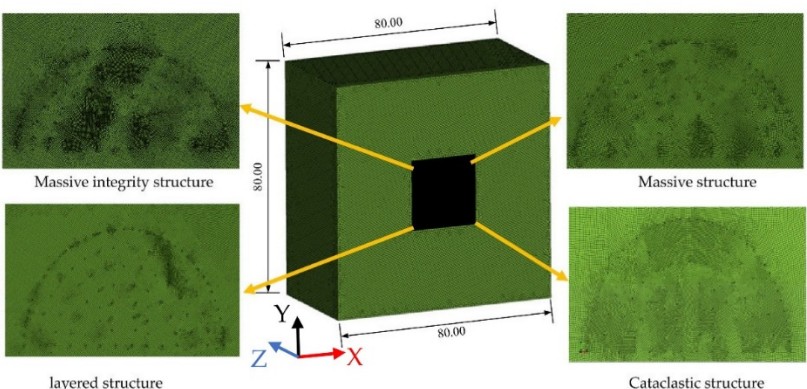

**Figure 8.** Finite element model (unit: m).

The properties of the structural plane were simulated by defining the normal and tangential stiffness of contact. Structural plane parameters referred to the formula of normal stiffness and tangential stiffness of rock mass structural plane proposed by Barton:

$$K_n = -7.15 + 1.75 JRC + 2(JCS/JRC), \tag{3}$$

$$K_S = 100/L \cdot JCS \cdot \tan \varphi_r, \tag{4}$$

where, $JRC$ is the roughness coefficient of the discontinuity, $JCS$ is the compressive strength of the discontinuity, $L$ is the trace length of the discontinuity, and $\varphi_r$ is the friction angle of rock mass. According to the field measurement, $JRC$ = 8, $JCS$ = 27.34 MPa, $L$ = 13 m, $\varphi_r$ = 57° were put into the model correction inversion, and the normal stiffness and tangential stiffness of the structural plane in the model were 13.685 GPa and 5.474 MPa, respectively.

In accordance with the actual situation of on-site blasting, the blast holes were arranged in the finite element model, the explosive center point was used as the detonation point, and the segmented micro-difference initiation technology was adopted. With reference to the actual situation of on-site blasting, blast holes were arranged in the finite element model. Taking the explosive center point as the initiation point, the piecewise millisecond initiation technology was adopted, and the initiation time difference of each hole in the model was set as 100 $\mu$s. Among them, the cut hole was detonated at 0 $\mu$s; the loosening hole was detonated at 100 $\mu$s; the satellite hole and the heading hole were detonated at 200 and 300 $\mu$s, respectively; and the peripheral hole was detonated at 400 $\mu$s.

The total computation time was 1000 μs. Solid 164 element was used for surrounding rock and explosives, and the numerical model created 2,504,199 nodes and 2,407,428 elements.

　　　The tunnel contours with different rock mass structures after blasting are shown in Figure 9. The influence of rock mass structure effect on tunnel smooth blasting quality was obvious. The numerical simulation of the four monitoring sections was compared with the on-site monitoring of the overbreak and underbreak of the tunnel face. Figure 10 shows the profile comparison of each section design, field monitoring, and numerical simulation. It can be seen that the design, field monitoring, and numerical simulation contours were similar in shape, and the overbreak and underbreak positions of the tunnel contour in numerical simulation and field monitoring were also relatively consistent. The overbreak of the tunnel was obvious, and the amount of underbreak was very small.

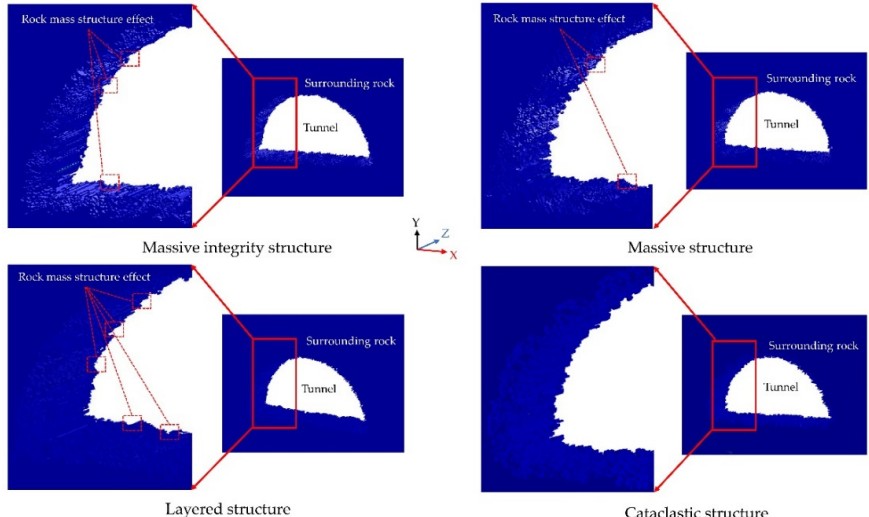

**Figure 9.** Tunnel contours with different rock mass structures.

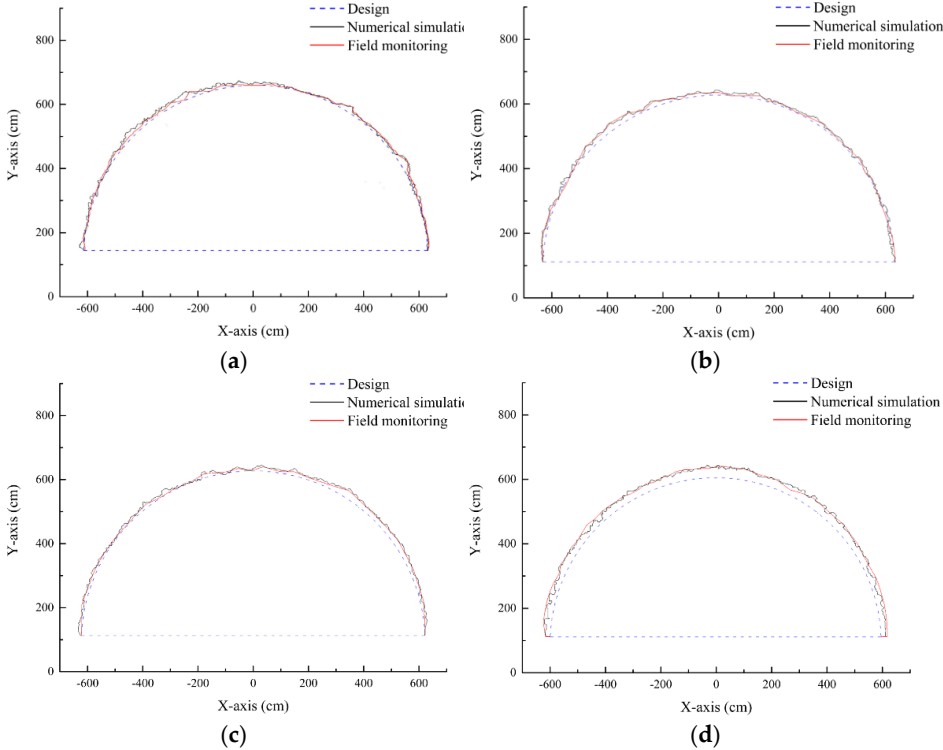

**Figure 10.** Comparison of section contour between numerical simulation and field monitoring: (**a**) massive integrity structure; (**b**) massive structure; (**c**) layered structure; and (**d**) cataclastic structure.

Table 3 shows the tunnel smooth blasting section area of numerical simulation and field monitoring. The designed tunnel section area was 63.48 m². For the massive integrity structure rock mass, the tunnel section area of the field test was 65.81 m², and the section overbreak rate was 3.67%. The tunnel section area of the numerical simulation was 65.94 m² and the overbreak rate was 3.88%. For the massive structure rock mass, the tunnel section contour area of field test was 66.45 m², and the section overbreak rate was 4.68%. The numerical simulation tunnel section contour area was 66.59 m², and the section overbreak rate was 4.88%. For the layered structure rock mass, the tunnel section contour area of field test was 66.95 m², and the section overbreak rate was 5.47%. The numerical simulation tunnel section contour area was 66.85 m², and the section overbreak rate was 5.32%.

**Table 3.** Tunnel smooth blasting section area of numerical simulation and field monitoring.

| Rock Structure Type | Area (m²) | | |
| --- | --- | --- | --- |
| | Field Monitoring | Numerical Simulation | Different |
| Massive integrity structure | 65.81 | 65.94 | 0.19% |
| Massive structure | 66.45 | 66.59 | 0.21% |
| Layered structure | 66.95 | 66.85 | 0.15% |
| Cataclastic structure | 71.51 | 71.63 | 0.17% |

For the cataclastic structure rock mass, the tunnel section contour area of field test was 71.51 m², and the section overbreak rate was 12.65%. The numerical simulation tunnel section contour area was 71.63 m², and the section overbreak rate was 12.84%. Comparing the overbreak rate of different rock mass structure, the overbreak rate increased with the decrease in the integrity of rock mass structure. It also can be seen that the difference between the tunnel smooth blasting section area of numerical simulation and field monitoring was very small, indicating that the numerical model adopted in this paper had high reliability.

*2.5. Orthogonal Numerical Test Design of Rock Mass Structure Effect*

In smooth blasting, many factors affect the overbreak and underbreak of the tunnel, mainly including those related to blast hole layout scheme and rock mass structure, such as peripheral hole spacing, heading hole spacing, the vertical distance between heading hole and peripheral hole, rock mass density, compressive strength, other physical and mechanical parameters, and structural plane distribution. All have varying degrees of influence on the blasting effect. If the influence of various factors on the overbreak and underbreak of tunnel smooth blasting was studied through comprehensive numerical tests, the number could be huge. Orthogonal test design is a test design method that uses the orthogonal table to arrange and analyze multifactor tests. It is an efficient test design method to study the influence of multi factors and multi levels through the orthogonal table. It also uses some tests to replace all tests to reflect the overall situation. Therefore, in this study, an orthogonal numerical test scheme was designed to comprehensively analyze the influencing factors of the overbreak and underbreak of smooth blasting in tunnels with different rock mass structures. Due to the differences in the influencing factors of overbreak and underbreak of tunnel smooth blasting in different rock mass structures, different influencing factors were considered for different rock mass structures. The orthogonal numerical test factors of the massive integrity structure, massive structure, layered structure, and cataclastic structure rock mass are shown in Tables 4–7.

**Table 4.** The influencing factors and level of the massive integrity structure rock mass.

| Level | Factors | | | | | |
|---|---|---|---|---|---|---|
| | $\gamma$ (°) | $\alpha$ (°) | $D$ (m) | $P$ (cm) | $H$ (cm) | $S$ (cm) |
| 1 | 0 | 0 | 3.0 | 40 | 90 | 40 |
| 2 | 20 | 20 | 3.5 | 50 | 100 | 50 |
| 3 | 40 | 40 | 4.0 | 60 | 110 | 60 |
| 4 | 60 | 60 | 4.5 | 70 | 120 | 70 |
| 5 | 80 | 80 | 5.0 | 80 | 130 | 80 |

**Table 5.** The influencing factors and level of the massive structure rock mass.

| Level | Factors | | | | | |
|---|---|---|---|---|---|---|
| | $\rho$ (g/cm$^3$) | $\theta$ (°) | $D$ (m) | $P$ (cm) | $H$ (cm) | $S$ (cm) |
| 1 | 1.5 | 0 | 3.0 | 40 | 90 | 40 |
| 2 | 1.8 | 20 | 3.5 | 50 | 100 | 50 |
| 3 | 2.1 | 40 | 4.0 | 60 | 110 | 60 |
| 4 | 2.4 | 60 | 4.5 | 70 | 120 | 70 |
| 5 | 2.7 | 80 | 5.0 | 80 | 130 | 80 |

**Table 6.** The influencing factors and level of the layered structure rock mass.

| Level | Factors | | | | | |
|---|---|---|---|---|---|---|
| | $\gamma$ (°) | $\alpha$ (°) | $D$ (m) | $P$ (cm) | $H$ (cm) | $S$ (cm) |
| 1 | 0 | 0 | 1.0 | 40 | 90 | 40 |
| 2 | 20 | 20 | 1.25 | 50 | 100 | 50 |
| 3 | 40 | 40 | 1.5 | 60 | 110 | 60 |
| 4 | 60 | 60 | 1.75 | 70 | 120 | 70 |
| 5 | 80 | 80 | 1.8 | 80 | 130 | 80 |

**Table 7.** The influencing factors and level of the cataclastic structure rock mass.

| Level | Factors | | | | | |
|---|---|---|---|---|---|---|
| | $\rho$ (g/cm$^3$) | $\sigma_c$ (MPa) | $Q$ | $P$ (cm) | $H$ (cm) | $S$ (cm) |
| 1 | 1.5 | 50 | 0.2 | 40 | 90 | 40 |
| 2 | 1.8 | 70 | 0.4 | 50 | 100 | 50 |
| 3 | 2.1 | 90 | 0.6 | 60 | 110 | 60 |
| 4 | 2.4 | 110 | 0.8 | 70 | 120 | 70 |
| 5 | 2.7 | 130 | 1.0 | 80 | 130 | 80 |

The orthogonal numerical test design in the massive integrity structure rock mass adopted the L$_{25}$ (5$^6$) orthogonal table. The factors and levels of combination design of each test group are shown in Table 8. The L$_{25}$ (5$^6$) orthogonal table was also used for the orthogonal numerical test design of the other three rock mass structures.

**Table 8.** Orthogonal numerical test design table of massive integrity structure rock mass.

| Test No. | Factors | | | | | |
|---|---|---|---|---|---|---|
| | $\gamma$ (°) | $\alpha$ (°) | $D$ (m) | $P$ (cm) | $H$ (cm) | $S$ (cm) |
| No. 1 | 1 (0) | 1 (0) | 1 (3.0) | 1 (40) | 1 (90) | 1 (40) |
| No. 2 | 1 (0) | 2 (20) | 4 (4.5) | 5 (80) | 2 (100) | 4 (70) |
| No. 3 | 1 (0) | 3 (40) | 2 (3.5) | 4 (70) | 3 (110) | 2 (50) |
| No. 4 | 1 (0) | 4 (60) | 5 (5.0) | 3 (60) | 4 (120) | 5 (80) |
| No. 5 | 1 (0) | 5 (80) | 3 (4.0) | 2 (50) | 5 (130) | 3 (60) |

**Table 8.** *Cont.*

| Test No. | Factors | | | | | |
|---|---|---|---|---|---|---|
| | $\gamma$ (°) | $\alpha$ (°) | *D* (m) | *P* (cm) | *H* (cm) | *S* (cm) |
| No. 6 | 2 (20) | 1 (0) | 3 (4.0) | 4 (70) | 4 (120) | 4 (70) |
| No. 7 | 2 (20) | 2 (20) | 1 (3.0) | 3 (60) | 5 (130) | 2 (50) |
| No. 8 | 2 (20) | 3 (40) | 4 (4.5) | 2 (50) | 1 (90) | 5 (80) |
| No. 9 | 2 (20) | 4 (60) | 2 (3.5) | 1 (40) | 2 (100) | 3 (60) |
| No. 10 | 2 (20) | 5 (80) | 5 (5.0) | 5 (80) | 3 (110) | 1 (40) |
| No. 11 | 3 (40) | 1 (0) | 5 (5.0) | 2 (50) | 2 (100) | 2 (50) |
| No. 12 | 3 (40) | 2 (20) | 3 (4.0) | 1 (40) | 3 (110) | 5 (80) |
| No. 13 | 3 (40) | 3 (40) | 1 (3.0) | 5 (80) | 4 (120) | 3 (60) |
| No. 14 | 3 (40) | 4 (60) | 4 (4.5) | 4 (70) | 5 (130) | 1 (40) |
| No. 15 | 3 (40) | 5 (80) | 2 (3.5) | 3 (60) | 1 (90) | 4 (70) |
| No. 16 | 4 (60) | 1 (0) | 2 (3.5) | 5 (80) | 5 (130) | 5 (80) |
| No. 17 | 4 (60) | 2 (20) | 5 (5.0) | 4 (70) | 1 (90) | 3 (60) |
| No. 18 | 4 (60) | 3 (40) | 3 (4.0) | 3 (60) | 2 (100) | 1 (40) |
| No. 19 | 4 (60) | 4 (60) | 1 (3.0) | 2 (50) | 3 (110) | 4 (70) |
| No. 20 | 4 (60) | 5 (80) | 4 (4.5) | 1 (40) | 4 (120) | 2 (50) |
| No. 21 | 5 (80) | 1 (0) | 4 (4.5) | 3 (60) | 3 (110) | 3 (60) |
| No. 22 | 5 (80) | 2 (20) | 2 (3.5) | 2 (50) | 4 (120) | 1 (40) |
| No. 23 | 5 (80) | 3 (40) | 5 (5.0) | 1 (40) | 5 (130) | 4 (70) |
| No. 24 | 5 (80) | 4 (60) | 3 (4.0) | 5 (80) | 1 (90) | 2 (50) |
| No. 25 | 5 (80) | 5 (80) | 1 (3.0) | 4 (70) | 2 (100) | 5 (80) |

## 3. Results and Discussions

The influence of various factors on the quality of tunnel smooth blasting was quantitatively analyzed using a single factor with the overbreak and underbreak rate of tunnel smooth blasting (the ratio of overbreak and underbreak area of tunnel section to design area). On the basis of the theory of fuzzy mathematics, the fuzzy orthogonal analysis method was introduced to analyze the main effect of the test results to study the influence and function of various factors. The construction steps of the fuzzy orthogonal analysis refer to the literature [35].

### 3.1. Massive Integrity Structure Rock Mass

The overbreak or underbreak rate of smooth blasting under different levels of each factor was sorted, and the single factor influence test results of each factor were obtained, as shown in Figure 11.

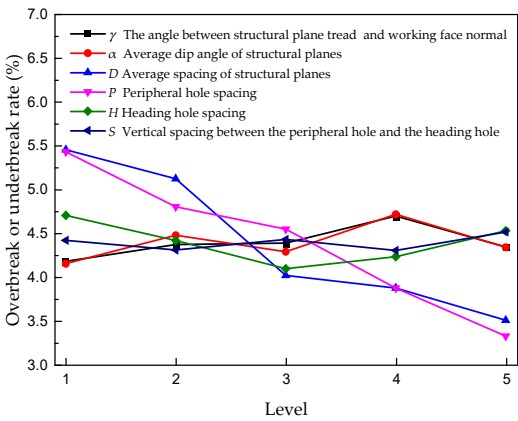

**Figure 11.** Overbreak or underbreak rate of smooth blasting under different levels of each factor in the massive integrity structure rock mass.

Among the six test factors, the peripheral hole spacing had the most significant impact on the smooth blasting overbreak rate, and the overbreak percentage decreased steadily

with the increase in peripheral hole spacing. When the peripheral hole spacing increased from 40 cm to 80 cm, the overbreak rate decreased from 5.43% to 3.32%. The influence of the average spacing of structural planes was second only to the peripheral hole spacing. When the average spacing of structural planes was 3.0 m, the overbreak rate was 5.45%, and when the average spacing of structural planes increased to 4.0 m, the overbreak rate decreased to 4.01%, with a decrease of 35.91%. Thus, the average spacing of structural planes had a more obvious influence on the overbreak rate. The smaller the average spacing of structural planes was, the denser the structural planes, and the greater the overbreak rate. However, when the average spacing of structural planes was increased from 4.0 m to 5.0 m, the overbreak rate dropped from 4.01% to 3.50%, which was only 14.57%, and the decline of the curve greatly slowed down. When the average spacing of structural planes was greater than 4.0 m, its influence on the overbreak rate of smooth blasting was small. The influence of the angle between the structural plane trend and working face normal ranked third. The overbreak rate increased slightly with the increase in the angle between the structural plane trend and working face normal. When the angle between the structural plane trend and working face normal was 60°, the overbreak rate was the largest. The overbreak rate increased with the increase in the dip angle of the structural plane, but the variation range was very small. The influence of the dip angle of the structural plane on the overbreak rate was much less than that of the average spacing of structural planes. The heading hole spacing and vertical spacing between the peripheral hole and the heading hole had a slight influence on the overbreak rate, and the curves were generally stable, which were not the main control factors of overbreak rate.

Taking the overbreak rate of smooth blasting as the research object, the membership value of each factor index and the fuzzy comprehensive evaluation value are shown in Table 9.

According to the main effect analysis of fuzzy mathematics theory, the maximum membership degree of the angle between structural plane trend and working face normal, average dip angle of the structural plane, average spacing of structural planes, peripheral hole spacing, heading hole spacing, the vertical spacing between the peripheral hole and the heading hole was $\hat{F}_\gamma = 0.223$, $\hat{F}_\alpha = 0.215$, $\hat{F}_D = 0.257$, $\hat{F}_P = 0.258$, $\hat{F}_H = 0.215$, and $\hat{F}_S = 0.204$, respectively. The maximum membership degree represented the influence of the specified factor on the evaluation index, and the value ranges from 0 to 1. The larger the value, the more obvious the influence of the factor on the evaluation index. Considering that $\hat{F}_P > \hat{F}_D > \hat{F}_\gamma > \hat{F}_H > \hat{F}_\alpha > \hat{F}_S$, the order of the influence weights of these six factors was as follows: peripheral hole spacing, average spacing of structural planes, the angle between structural plane trend and working face normal, heading hole spacing, average dip angle of the structural plane, and vertical spacing between the peripheral hole and the heading hole. The peripheral hole spacing was the most influential factor with the weighted factor, and its maximum membership degree was 0.258, much greater than the maximum membership degree of the heading hole spacing and vertical spacing between the peripheral hole and the heading hole. For the massive integrity structure rock mass, the influence of the peripheral hole spacing on the overbreak rate of smooth blasting was much greater than that of the layout spacing of the blast holes inside the section, and the peripheral hole spacing should be paid special attention to. In addition, the maximum degree of membership of the average spacing of structural planes was second only to the peripheral hole spacing, reaching 0.257, indicating that the spacing between structural planes also had a great influence on the overbreak rate.

**Table 9.** Membership degree and fuzzy evaluation value of the overbreak rate of smooth blasting of the massive integrity structure rock mass.

| Test No. | Influence Factor | | | | | | Results | |
|---|---|---|---|---|---|---|---|---|
| | $\gamma$ (°) | $\alpha$ (°) | $D$ (m) | $P$ (cm) | $H$ (cm) | $S$ (cm) | Overbreak Rate (%) | $r_{ij}$ |
| No. 1 | 1 (0) | 1 (0) | 1 (3.0) | 1 (40) | 1 (90) | 1 (40) | 6.36 | 0.74 |
| No. 2 | 1 (0) | 2 (20) | 4 (4.5) | 5 (80) | 2 (100) | 4 (70) | 2.59 | 0.30 |
| No. 3 | 1 (0) | 3 (40) | 2 (3.5) | 4 (70) | 3 (110) | 2 (50) | 3.89 | 0.45 |
| No. 4 | 1 (0) | 4 (60) | 5 (5.0) | 3 (60) | 4 (120) | 5 (80) | 3.71 | 0.43 |
| No. 5 | 1 (0) | 5 (80) | 3 (4.0) | 2 (50) | 5 (130) | 3 (60) | 4.32 | 0.50 |
| No. 6 | 2 (20) | 1 (0) | 3 (4.0) | 4 (70) | 4 (120) | 4 (70) | 2.97 | 0.35 |
| No. 7 | 2 (20) | 2 (20) | 1 (3.0) | 3 (60) | 5 (130) | 2 (50) | 5.71 | 0.66 |
| No. 8 | 2 (20) | 3 (40) | 4 (4.5) | 2 (50) | 1 (90) | 5 (80) | 4.58 | 0.53 |
| No. 9 | 2 (20) | 4 (60) | 2 (3.5) | 1 (40) | 2 (100) | 3 (60) | 6.51 | 0.76 |
| No.10 | 2 (20) | 5 (80) | 5 (5.0) | 5 (80) | 3 (110) | 1 (40) | 2.07 | 0.24 |
| No. 11 | 3 (40) | 1 (0) | 5 (5.0) | 2 (50) | 2 (100) | 2 (50) | 3.60 | 0.42 |
| No. 12 | 3 (40) | 2 (20) | 3 (4.0) | 1 (40) | 3 (110) | 5 (80) | 4.94 | 0.58 |
| No. 13 | 3 (40) | 3 (40) | 1 (3.0) | 5 (80) | 4 (120) | 3 (60) | 4.14 | 0.48 |
| No. 14 | 3 (40) | 4 (60) | 4 (4.5) | 4 (70) | 5 (130) | 1 (40) | 3.82 | 0.44 |
| No. 15 | 3 (40) | 5 (80) | 2 (3.5) | 3 (60) | 1 (90) | 4 (70) | 5.43 | 0.63 |
| No. 16 | 4 (60) | 1 (0) | 2 (3.5) | 5 (80) | 5 (130) | 5 (80) | 4.36 | 0.51 |
| No. 17 | 4 (60) | 2 (20) | 5 (5.0) | 4 (70) | 1 (90) | 3 (60) | 3.71 | 0.43 |
| No. 18 | 4 (60) | 3 (40) | 3 (4.0) | 3 (60) | 2 (100) | 1 (40) | 4.40 | 0.51 |
| No. 19 | 4 (60) | 4 (60) | 1 (3.0) | 2 (50) | 3 (110) | 4 (70) | 6.09 | 0.71 |
| No. 20 | 4 (60) | 5 (80) | 4 (4.5) | 1 (40) | 4 (120) | 2 (50) | 4.90 | 0.57 |
| No. 21 | 5 (80) | 1 (0) | 4 (4.5) | 3 (60) | 3 (110) | 3 (60) | 3.46 | 0.40 |
| No. 22 | 5 (80) | 2 (20) | 2 (3.5) | 2 (50) | 4 (120) | 1 (40) | 5.42 | 0.63 |
| No. 23 | 5 (80) | 3 (40) | 5 (5.0) | 1 (40) | 5 (130) | 4 (70) | 4.42 | 0.52 |
| No. 24 | 5 (80) | 4 (60) | 3 (4.0) | 5 (80) | 1 (90) | 2 (50) | 3.44 | 0.40 |
| No. 25 | 5 (80) | 5 (80) | 1 (3.0) | 4 (70) | 2 (100) | 5 (80) | 4.96 | 0.58 |
| $\sum b_{i1}$ | 4.17 | 4.15 | 5.45 | 5.43 | 4.70 | 4.42 | | |
| $(\sum b_{i1})$ | 0.190 | 0.189 | 0.257 | 0.258 | 0.215 | 0.201 | | |
| $\sum b_{i2}$ | 4.37 | 4.47 | 5.12 | 4.80 | 4.41 | 4.31 | | |
| $(\sum b_{i2})$ | 0.199 | 0.204 | 0.233 | 0.219 | 0.201 | 0.196 | | |
| $\sum b_{i3}$ | 4.38 | 4.29 | 4.01 | 4.54 | 4.09 | 4.43 | | |
| $(\sum b_{i3})$ | 0.200 | 0.195 | 0.183 | 0.207 | 0.186 | 0.202 | | |
| $\sum b_{i4}$ | 4.69 | 4.71 | 3.87 | 3.87 | 4.23 | 4.30 | | |
| $(\sum b_{i4})$ | 0.223 | 0.215 | 0.176 | 0.176 | 0.193 | 0.196 | | |
| $\sum b_{i5}$ | 4.34 | 4.34 | 3.50 | 3.32 | 4.53 | 4.51 | | |
| $(\sum b_{i5})$ | 0.198 | 0.197 | 0.159 | 0.151 | 0.206 | 0.204 | | |

*3.2. Massive Structure Rock Mass*

Through single factor analysis, the influence curves of various factors in the massive structure rock mass could be obtained, as shown in Figure 12. Among the six numerical test factors, the angle between structural planes had the most significant influence on the overbreak rate of smooth blasting, and the overbreak rate decreased rapidly with the increase in the angle between the structural planes. When the angle between the structural planes increased from 30° to 90°, the overbreak rate decreased from 6.96% to 3.46%. The slope of the influence curve of the average spacing of structural planes was obviously smaller than that of the angle between structural planes, indicating that the influence of the average spacing of structural planes was second only to the angle between structural planes. For massive structural rock mass, the smaller the angle between structural planes was, the more serious the fragmentation degree of rock mass was. Furthermore, the influence of the angle between structural planes on the fragmentation degree was greater than that of the spacing between structural planes. The influence of peripheral hole spacing was the third, and the smooth blasting overbreak rate decreased slowly with the increase in peripheral hole spacing. The curves of rock density, heading hole spacing, and vertical

spacing between the peripheral hole and the heading hole were relatively stable on the whole. They exhibited a relatively slight influence on the overbreak rate, hence not the main control factors of overbreak rate, and have relatively slight influence on the overbreak rate, which were not the main control factors of overbreak rate.

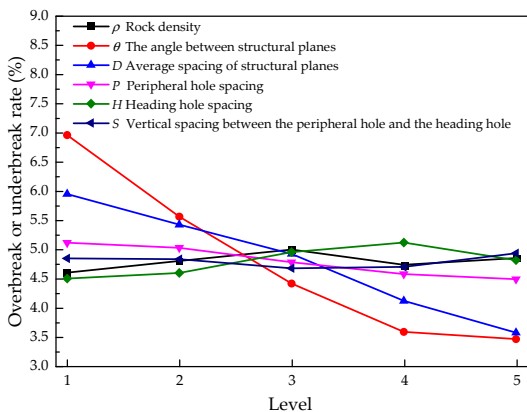

**Figure 12.** Overbreak or underbreak rate of smooth blasting under different levels of each factor in the massive structure rock mass.

Through the main effect analysis, the maximum membership degree of rock density, the angle between structural planes, the average spacing of structural planes, the peripheral hole spacing, the heading hole spacing, and the vertical spacing between the peripheral hole and the heading hole were obtained, where $\hat{F}_\rho = 0.208$, $\hat{F}_\theta = 0.287$, $\hat{F}_D = 0.246$, $\hat{F}_P = 0.222$, $\hat{F}_H = 0.207$, and $\hat{F}_S = 0.206$, respectively. Considering $\hat{F}_\theta > \hat{F}_D > \hat{F}_P > \hat{F}_\rho > \hat{F}_H > \hat{F}_S$, the influence weight of these six factors were ordered as the angle between structural planes, average spacing of structural planes, peripheral hole spacing, rock density, heading hole spacing, and vertical spacing between the peripheral hole and the heading hole. The maximum membership degree of the angle between structural planes was 0.287, much higher than the maximum membership degree of other factors, indicating that the angle between structural planes was the biggest influencing factor of the weight. The influence degree of the average spacing of structural planes was the second, and its maximum membership degree was 0.246. Due to the cutting of two groups of structural planes, the smaller the angle between the two groups of structural planes was, the smaller the spacing of structural planes, the more fragmented the rock mass, and the more serious the overbreak was. The peripheral hole spacing with the third largest membership was only 0.222, which was far less than the former two, indicating that the influence of rock mass structure on massive rock mass was far greater than that of blast hole layout.

### 3.3. Layered Structure Rock Mass

Figure 13 shows the influence curve of various factors of the layered rock mass. Among the six numerical test factors, the average spacing of structural planes had the most significant influence on the overbreak rate of smooth blasting, and the overbreak percentage decreased rapidly with the increase in the average spacing of structural planes. When the average structural planes spacing increased from 1.0 m to 2.0 m, the overbreak rate decreased from 8.11% to 3.65%. The influence of peripheral hole spacing was second only to the average spacing of structural planes. When the peripheral hole spacing was 40 cm, the overbreak rate was 6.89%, and when the peripheral hole spacing increased to 80 cm, the overbreak rate decreased to 4.90%. When the peripheral hole spacing increased by 10 cm, the overbreak rate decreased by 0.71%. The influence of the angle between structural plane trend and working face normal ranked third. With the increase in $\gamma$, the overbreak rate also increased. When the angle was 60°, the overbreak rate was the largest.

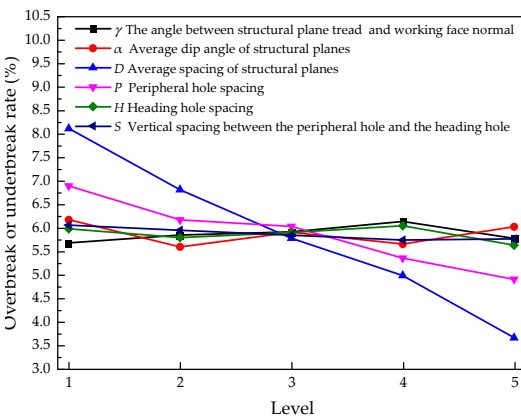

**Figure 13.** Overbreak or underbreak rate of smooth blasting under different levels of each factor in the layered structure rock mass.

Through the main effect analysis, the maximum membership degree of the angle between structural plane trend and working face normal, the average dip angle of the structural plane, the average spacing of structural planes, the peripheral hole spacing, the heading hole spacing, and the vertical spacing between the peripheral hole and the heading hole were obtained as follows: $\hat{F}_\gamma = 0.209, \hat{F}_\alpha = 0.213$, $\hat{F}_D = 0.284$, $\hat{F}_P = 0.244$, $\hat{F}_H = 0.205$, and $\hat{F}_S = 0.205$, respectively. Considering that $\hat{F}_D > \hat{F}_P > \hat{F}_\alpha > \hat{F}_\gamma > \hat{F}_H = \hat{F}_S$, the influence weight of these six factors was ordered as the average spacing of structural planes, peripheral hole spacing, average dip angle of the structural plane, the angle between structural plane trend and working face normal, heading hole spacing, and vertical spacing between the peripheral hole and the heading hole. The maximum membership degree of the average spacing of structural planes reached 0.284, which was much greater than that of other factors, indicating that the average spacing of structural planes was the most influential factor. The results showed that the smaller the structural plane spacing of the layered structure was, the greater the degree of rock mass fragmentation, and the more serious the overbreak of smooth blasting was.

### 3.4. Cataclastic Structure Rock Mass

Through single factor analysis, the influence curves of various factors of cataclastic structure could be obtained, as shown in Figure 14. The peripheral hole spacing had the most significant effect on the overbreak rate of smooth blasting. When the peripheral hole spacing increased from 40 cm to 50 cm, the overbreak rate of the tunnel decreased from 14.98% to 10.07%, with a decrease of 32.77%. However, when the peripheral hole spacing increased from 50 cm to 80 cm, the overbreak rate decreased by only 4.68%, which reflected that the overbreak rate decreased with the increasing peripheral eye spacing, and the decrease was slower. When the peripheral hole spacing was greater than 50 cm, its influence on the overbreak rate was relatively small. For cataclastic structure rock mass, the overbreak rate was generally serious when the peripheral hole spacing was less than 50 cm. Thus, the peripheral hole spacing should not be less than 50 cm. When the rock density increased from 1.5 $g/cm^3$ to 1.8 $g/cm^3$, the overbreak rate decreased from 14.91% to 14.27%, with a decrease of only 4.09%. However, when the rock density increased from 1.8 $g/cm^3$ to 2.1 $g/cm^3$ and from 2.1 $g/cm^3$ to 2.4 $g/cm^3$, the reduction of overbreak rate reached 23.57% and 45.39%, respectively. When the rock density was less than 1.8 $g/cm^3$, the density had a limited influence on the blasting overbreak rate. When the density was greater than 1.8 $g/cm^3$, the overbreak rate decreased rapidly as the density increased. Therefore, for cataclastic structure rock mass, the density of less than 2.0 $g/cm^3$ had a great influence on smooth blasting overbreak. Thus, special attention should be paid to blasting design and construction. The influence of the $Q$ classification index on smooth blasting overbreak was less than that of rock density. The curve showed that the overbreak rate decreased with the increase in $Q$, and the decreasing range was relatively stable. In the

cataclastic structure rock mass, the structural planes were extremely developed, and $Q$ was a characterization method of the development degree of the structural plane. The smaller the $Q$ was, the more fragmented the rock mass was, resulting in the overbreak rate increasing with the decrease in $Q$.

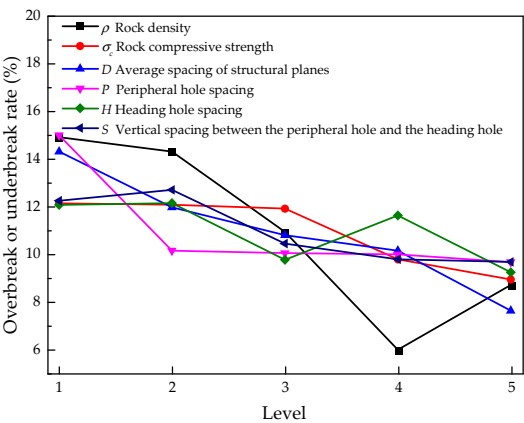

**Figure 14.** Overbreak or underbreak rate of smooth blasting under different levels of each factor in the cataclastic structure rock mass.

According to the main effect analysis of fuzzy mathematics theory, the maximum membership degree of the rock density, rock compressive strength, Q classification index, peripheral hole spacing, heading hole spacing, and the vertical spacing between the peripheral hole and the heading hole was $\hat{F}_{\rho} = 0.272$, $\hat{F}_{\sigma_c} = 0.221$, $\hat{F}_Q = 0.261$, $\hat{F}_P = 0.274$, $\hat{F}_H = 0.221$, and $\hat{F}_S = 0.232$, respectively. Given that $\hat{F}_P > \hat{F}_{\rho} > \hat{F}_Q > F_S > \hat{F}_{\sigma_c} > \hat{F}_H$, the order of the influence weights of these six factors was peripheral hole spacing, rock density, Q classification index, the vertical spacing between the peripheral hole and the heading hole, rock compressive strength, and heading hole spacing. The maximum degree of membership of the peripheral hole spacing was 0.274, and the peripheral hole spacing was the most influential factor. For cataclastic structure rock mass, to reduce the overbreak rate in smooth blasting, the peripheral hole spacing should be strictly controlled, and the peripheral hole spacing should not be less than 50 cm. The maximum membership degree of the $Q$ classification index was 0.261, with a large weight, indicating that the development degree of the structural plane had a great influence on smooth blasting. The more fragmented the rock mass was, the more prone it was to overbreak.

## 4. Conclusions

(1) With Zigaojian tunnel as background, the rock mass structure effect of smooth blasting quality in tunnels was analyzed. Onsite information collection, field monitoring, rock mass structure reconstruction in software, numerical verification and orthogonal numerical tests were performed to evaluate the rock mass structure effect.

(2) The rock mass structure effect influenced the smooth blasting quality of a tunnel obviously.

(3) The tunnel smooth blasting quality decreased with the decrease in rock mass integrity. The influencing factors of rock mass structure effect depended on the rock mass structure types.

(4) The peripheral hole spacing was the most important factor affecting the smooth blasting quality of massive integrity structure rock mass and cataclastic structure rock mass. Strictly controlling the peripheral hole spacing can reduce the overbreak rate of smooth blasting.

(5) The angle between structural planes was the most important factor affecting the smooth blasting quality of massive structure rock mass, and the average spacing of structural planes was the main factor affecting the smooth blasting quality of layered structure rock mass, which was much greater than the peripheral hole spacing.

(6) The rock mass structure effect of smooth blasting quality in the tunnel can provide a reference for the rock mass structure effect of other similar cases.

**Author Contributions:** Conceptualization, J.W.; funding acquisition, H.W.; investigation, X.L., H.L., and Y.S.; software, J.L.; writing – original draft, A.C. and J.W.; validation, Y.L. and F.W. All authors have read and agreed to the published version of the manuscript.

**Funding:** This research was funded by the Shanghai Municipal Science and Technology Project (18DZ1201301; 19DZ1200900); Xiamen Road and Bridge Group (XM2017-TZ0151; XM2017-TZ0117); the project of Key Laboratory of Impact and Safety Engineering (Ningbo University), Ministry of Education (CJ202101); Shanghai Municipal Science and Technology Major Project (2021SHZDZX0100) and the Fundamental Research Funds for the Central Universities; and the Key Laboratory of Land Subsidence Monitoring and Prevention, Ministry of Natural Resources of the People's Republic of China (No. KLLSMP202101), Suzhou Rail Transit Line 1 Co., Ltd., China Railway 15 Bureau Group Co., Ltd.

**Conflicts of Interest:** The authors declare no conflict of interest.

## Abbreviations

| | |
|---|---|
| PPV | Peak particle velocity |
| JHC | Johnson–Holmquist–Cook |
| $\gamma$ | Angle between structural plane trend and working face normal |
| $\alpha$ | Average dip angle of structural planes |
| $D$ | Average spacing of structural planes |
| $P$ | Peripheral hole spacing |
| $H$ | Heading hole spacing |
| $S$ | Vertical spacing between the peripheral hole and the heading hole |
| $\rho$ | Rock density |
| $\theta$ | Angle between structural planes |
| $\sigma_c$ | Rock compressive strength |
| $Q$ | $Q$ classification index |

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
