# Peer review of "Numerical Simulation of Rock Mass Structure Effect on Tunnel Smooth Blasting Quality: A Case Study"

_applsci, doi:10.3390/app112210761_

Round 1
Reviewer 1 Report
The authors investigated the influence of rock mass structure on the tunnel smooth blasting quality through numerical simulation. Generally speaking, it is a well written article. In my opinion, this article can be accepted for publication should the authors address the following concerns:
- The authors need to conduct a thorough proofreading before resubmitting. There are lots of mistakes in grammar.
- How do the authors simulate the structures in rock mass and how to determine their parameters?
- The authors are encouraged to give some simulation results (in pictures) regarding the tunnel shape after blasting.
- Do the behaviors of structures have influence on the overbreak or underbreak rate of smooth blasting?
Reviewer 2 Report
The concept of applying the described method in other cases is missing in the text or perhaps in the conclusions.
Reviewer 3 Report
Subject: Review, October 17, 2021
Applied science
Title: Numerical simulation of rock mass structure effect on tunnel 2 smooth blasting quality A case study
Comments:
- The abstract is not very clear and long, it is hard to make out what is being presented. No obvious terminology citing what is the problem being solved. It has to be re-written to identify the research work planned. In its present form, it sounds like a mini introduction.
- The authors should add a nomenclature to define the abbreviations and parameters used throughout the paper.
- Figure 2(c) which represent the geometric model. It would be helpful to identify the tool used to generate such model and adding dimensions is very useful to the reader.
- The 3D scanning tool used to generate the rock model should be also identified.
- As noted in item 3 above, the same applies to Figures 4 and 5.
- Figure 6 should have dimensional units.
- In the section “numerical Model and Verification”, the authors should define the finite element model more clearly, type of elements and number, number of nodes. Additionally, the boundary conditions applied can be highlighted on the model. This will give more value to modeling.
- Dimensional units are missing on Figure 8.
- On Figures 10 through 13, the authors should write out the name of the legend for each symbol shown on the graph.
- The numerical sections of the work presented in this paper is rather condense and hard to follow. The authors should consider revising these sections to much smaller paragraphs and focus on presenting the essential data related to the tests and analysis. The link that relates the simulation with the testing is missing. This applies primarily to sections 2.5 and 3.
- The authors should indicate what was the data acquisition process applied.
- The authors do not provide detailed description on how the graphical plots were generated. Interpretation of the results is not very clear.
- The conclusion is not well understood. A bullet style format citing what was done and found will serve the reader better. In its present format, it is only a listing of a set of statements. Also, citing what may have affected the outcome of the results if any exist should be included.
Overall, the paper needs work, in its present form, it is hard follow and not enough explanation is given in terms of identifying the results obtained and demonstrating the methodology applied. The problem being targeted to be solved is not well recognized. The paper needs major re-organization to allow presenting the results in a more coherent process.
The paper cannot be recommended for publication in its present form, a major re-write to allow better understanding of the work performed is needed.
Round 2
Reviewer 3 Report
The paper reads better, the authors to some extents have complied with the recommendations. However, one note is, the abstract remains relatively long and a little unclear, it can use some improvement.
The paper needs good grammatical workup. Upon addressing these minor issues, it can be released for publication.
Author Response
Dear Editors and Reviewers,
Thank you very much for your efficient comments and many efforts to help us improve our manuscript entitled “Numerical simulation of rock mass structure effect on tunnel smooth blasting quality: A case study”. We are grateful and appreciate for your so detailed comments on the manuscript, figures and tables. We have made careful revisions on the original manuscript. All changes made to the text are marked up using the “Track Changes” so that they may be easily identified.
Here below is our respond to the reviewer’s comments:
Response to Reviewer 3
Point 1: The paper reads better, the authors to some extents have complied with the recommendations. However, one note is, the abstract remains relatively long and a little unclear, it can use some improvement.
Respond: Thank you very much. It has been revised. The original Abstract has been streamlined from 268 words to 168 words to make it more concise. The repeated and unnecessary parts were all deleted. The revisions were marked up using the “Track Changes” function. See lines 12-31.
The revised Abstract is listed as follow:
Taking the Zigaojian tunnel, Hangzhou-Huangshan high-speed railway, China as background, the rock mass structure effect on smooth blasting quality was studied. Four rock mass structures were determined on the basis of the information collected on the tunnel site. Smooth blasting finite element models were established using LS-DYNA. The accuracy of the numerical calculation model was verified by comparing the overbreak and underbreak between the numerical simulation and monitoring. Orthogonal numerical test was used to study the rock mass structure effect through single factor and main effect analysis methods. With the decrease in rock mass integrity, the smooth blasting overbreak of tunnels with massive integrity structure, massive structure, layered structure, and cataclastic structure increased. For massive integrity structure and cataclastic structure, the peripheral hole spacing should be emphatically considered. Meanwhile, in massive structure and layered structure, the included angle and spacing of structural planes had a great influence on the smooth blasting quality. The research results could provide a reference to improve the quality of similar tunnel smooth blasting.
The revision manuscript is listed in the attachment. Please see the attachment.
The paper needs good grammatical workup. Upon addressing these minor issues, it can be released for publication.
Respond: Thank you very much. It has been revised. All the revisions were marked up using the “Track Changes” function. See lines 54,56,70,77,152,172,246,290-292,379,502.
The revision manuscript is listed in the attachment. Please see the attachment.
